# Is Heterogeneity Better? The Impact of Top Management Team Characteristics on Enterprise Innovation Performance

**DOI:** 10.3390/bs12060164

**Published:** 2022-05-26

**Authors:** Haihong Wang, Wenjun He, Yufan Yang

**Affiliations:** 1Department of Tourism Management, School of Business, Liaoning University, Shenyang 110036, China; wanghaihong0209@163.com; 2Department of Marketing and Tourism Management, School of Economics and Management, Wuhan University, Wuhan 430072, China; hewenjun_joe@163.com; 3School of Business Administration, Faculty of Business Administration, Southwestern University of Finance and Economics, Chengdu 611130, China

**Keywords:** top management team characteristics heterogeneity, enterprise innovation performance, GEM-listed enterprise, shallow-factor heterogeneity, deep-factor heterogeneity

## Abstract

Although the importance of the top management team’s characteristics has attracted research attention, its influence remains undetermined. This paper considers the influence of top management team characteristics’ heterogeneity on GEM-listed enterprises’ innovation performance. From the perspective of optimizing top management team human capital and improving enterprise innovation performance, this study analyzes 634 GEM-listed enterprises via regression analysis. The results show that the effect of shallow-factor heterogeneity of the top management team on enterprise innovation performance is not significant, but deep-factor heterogeneity has a negative effect. These findings indicate that deep-factor heterogeneity is more important than shallow-factor heterogeneity and is subject to closer attention. If a GEM-listed enterprise wants to set up a strong top management team that contributes to the improvement of enterprise innovation performance, it must examine the consistency of each member’s educational background and extend their tenure as long as possible without considering the gender ratio or age structure of the team.

## 1. Introduction

The role of the top management team in the innovation activities of modern enterprises cannot be ignored [1,2]. This is particularly true for GEM-listed enterprises (China established the GEM in 2009 to support its financing and strengthen its independent innovation ability) [3], in which innovation is the core element of their development [4]. These individuals directly participate in the formulation and implementation of such companies’ innovation decisions and influence the grasping of innovation opportunities and the direction of innovative investments, which in turn affect the firms’ innovation performance [2]. Therefore, the backgrounds of the top management team deeply influence their decisions and, in turn, enterprise performance. Hambrick was the first to identify this influence, termed upper echelons theory, in 1984 [5]. According to this theory, differences between top management team members in terms of their cognitive concepts, values, and social experience affect the enterprise’s output [6,7,8]. These differences in backgrounds are referred to as the heterogeneity of the top management team and manifest as differences in gender, age, tenure, and education level [9]. While abundant previous research has shown that heterogeneity of the top management team is typically significant, its influence on innovation performance has not been conclusively determined [10,11,12]. Thus, this paper examines the characteristics of executive teams in GEM-listed enterprises and divides top management team heterogeneity into shallow-factor heterogeneity (e.g., gender, age) and deep-factor heterogeneity (e.g., tenure and education level) to examine their effects on enterprise innovation performance and provide a theoretical basis for optimizing the human capital of senior management teams.

The remainder of this article is organized as follows. Next, we review prior research on the relationships among top management team characteristics, upper echelons theory, and enterprise innovation performance. Second, we develop our research hypotheses. We then use regression analysis to test our hypotheses and conclude by discussing the implications and limitations of our findings.

## 2. Theoretical Review

### 2.1. Upper Echelons Theory and the Effects of Top Management Team Characteristics

Upper echelons theory, as proposed by Hambrick and Mason, has become a cornerstone of top management team research [5,13]. This theory suggests that the top management team is composed of the individuals tasked with disclosing information to external stakeholders of the enterprise [5]. In general, whether from the perspective of the corporate strategic decision-making process or the impact on corporate performance, the importance of top management teams in enterprises has been fully affirmed. For instance, a significant body of research explores the influence of age, gender, education level, and other backgrounds of the top management team on executive behaviors and corporate performance [14,15,16,17]. Furthermore, researchers have added to and revised the upper echelons theory, arguing that the top management team comprises the individuals that are most deeply involved in decision-making and -execution, and have begun to pay more attention to the important role of the top management team in strategic decisions and performance improvements [18,19,20].

Extant studies have used various approaches to examine the backgrounds of different enterprises’ top management teams. The main methods involve questionnaires administered directly to the chief executive officers (CEOs) [21] as well as information obtained through companys’ annual reports or initial public offerings (IPO) reports [18]. Considering the availability and objective credibility of the relevant data, most scholars have opted to gain information on senior management team members from the annual reports of listed companies and other materials [22]. We use the last approach in this study.

### 2.2. Top Management Team Heterogeneity and Innovation Performance

As strategic decision-makers, members of the top management team can bring strategic resources that are important to the development of the enterprise, which is of great significance to enterprise performance [18]. Based on upper echelons theory, the heterogeneity of top management team characteristics has been highlighted as an important factor affecting the long-term operation of enterprises [23]. Such heterogeneity refers to team members’ individual backgrounds—specifically, differences in age, tenure, education level, and other backgrounds [24]. The influence of background heterogeneity has been conceptualized differently under different theoretical frameworks. For example, researchers following the perspective of information processing theory have proposed that the higher the degree of heterogeneity, the better the results, such as a higher level of organizational learning and innovation in enterprises [25]. Specifically, some researchers have demonstrated that top management teams with a diverse knowledge base and career experience can more effectively make decisions in dynamic and changing environments and are more creative and able to solve specific problems, therefore improving innovation performance [26]. This is because the existence of heterogeneity means that members of the top management team can access different information sources and divergent views on the basis of differentiated cognition to solve problems, which is conducive to enterprise learning and performance improvement [24,27]. Other scholars have applied social identity theory to argue that individuals tend to associate more with those who are similar to them, such that heterogeneity will affect top management team members’ identity; inhibit the transfer of professional skills within the team [28,29]; increase the likelihood of conflicts; and produce greater differences in the process of formulating strategies, strategic objectives, and strategic plans [30,31]. However, studies on the impact of heterogeneity in top management teams have either treated the various backgrounds (gender, age, education background, tenure, etc.) of the top management team as a whole [32] or selected only one background (e.g., gender) as representative of the whole. The former approach fails to differentiate between the different background types, while the latter is overly homogeneous and leads to underrepresentation.

To better explain the divergent findings, top management team backgrounds have been divided into shallow factors, which include some factors that are predetermined by birth and relatively difficult to change, such as gender, race, age, nationality, etc.; and deep factors, including career experience, education background, industry status, tenure, and other variables that are more dependent on acquired experience [33]. Specifically, shallow factors in team members refer to differences based on some demographic characteristics variables, while deep factors are formed by the past learning and work experiences of team members [34]. Based on this division, scholars have been able to draw more valuable conclusions. For instance, differences in shallow factors, such as gender and age, have been found to lead to less patience for cooperation and less time for mutual understanding among team members, which may increase conflicts [35]; however, deep-factor diversity among team members brings more knowledge resources and perspectives, which have a positive impact on innovation [36]. Nevertheless, findings have been inconclusive and often conflicting across studies, with some authors arguing that differences in shallow factors have a positive influence [5] and others that deep-factor heterogeneity has a negative influence [37].

## 3. Research Hypotheses

Extant research has extensively studied the superficial factors of managers that are inherent and difficult to change (e.g., gender and age) [17,38,39]. Some researchers have suggested that these so-called shallow factors do not cause cognitive conflicts to the same extent that factors such as race do [40]. GEM companies are more dynamic, passionate, and inclusive than other companies with longer histories and larger structures [41], so there may be fewer gender stereotypes among top management teams. Thus, based on information processing theory, increased gender heterogeneity is conducive to promoting the decision-making and management efficiency of teams [25]. This is because it enlarges the vision of the top management team with regard to information collection, provides richer solutions, and reduces the need for excessive investment in enterprises. Moreover, due to the differences in cognitive perspectives and values between the genders, a top management team with a high degree of gender heterogeneity can incorporate more perspectives and deeper thinking when making innovative decisions, making it more likely to bring improvements to enterprise innovation performance [33]. The flexible nature of GEM companies is also a testament to the fact that they have not yet developed a strictly seniority-based corporate culture; therefore, under normal circumstances, young executives in the top management team tend to be creative and open to taking risks, while older managers have significant experience and a large interpersonal network, which can be well-integrated. Under such conditions, age heterogeneity means a wider range of intellectual assets, richer information sources, and a stronger ability to solve new problems compared to the case in more homogeneous teams [42]. Thus, top management teams with a high level of age heterogeneity have more diverse problem analysis perspectives and can provide more feasible solutions for enterprise innovation, which is conducive to improving firms’ innovation performance [5]. Therefore, the following hypotheses are proposed:

**Hypothesis** **1** **(H1).***Shallow-factor heterogeneity within the top management team positively affects GEM enterprises’ innovation performance*.

**Hypothesis** **1A** **(H1A).***Gender heterogeneity within the top management team positively affects GEM enterprises’ innovation performance*.

**Hypothesis** **1B** **(H1B).***Age heterogeneity within the top management team has a positive influence on GEM enterprises’ innovation performance*.

With respect to deep factors, variations in tenure and education levels affect the cognitive styles and values of senior executives [33]. Good communication between members is based on tenure length [43]; however, large differences in tenure reduce team cohesion and decision-making efficiency and have a negative influence on enterprise value [37,44]. Managers with short tenure tend to be stubborn and lack the ability to think about the big picture, resulting in frequent conflicts and difficulty reaching consensus with long-tenured managers, which is not conducive to the improvement of enterprise innovation value [37]. Thus, lower tenure heterogeneity can help top management team members become better acquainted and thereby increase tacit understanding, reduce conflict, and stimulate positive outcomes for the enterprise [44]. In terms of educational background, variations therein shape different mindsets; members of top management teams see themselves as having been successful in their careers (i.e., having reached the executive level) based on their own mindset and are therefore more likely to believe in their own methodologies and stick to them [30]. Thus, high education background heterogeneity will increase conflicts, make it difficult to reach a consensus on the company’s strategic decisions, reduce the decision-making efficiency of the team, and have a negative impact on the firm’s innovation performance. Therefore, the following additional hypotheses are proposed:

**Hypothesis** **2** **(H2).***Deep-factor heterogeneity within the top management team negatively affects enterprise innovation performance*.

**Hypothesis** **2A** **(H2A).***Tenure heterogeneity within the top management team negatively affects enterprise innovation performance*.

**Hypothesis** **2B** **(H2B).***Education background heterogeneity within the top management team negatively affects enterprise innovation performance*.

## 4. Method

### 4.1. Sample Selection and Data Sources

For GEM-listed enterprises, strong innovation performance is key to maintaining competitiveness and achieving sustainable development. China established the GEM in 2009 to support its financing and strengthen its independent innovation ability [3]. This paper takes GEM-listed enterprises as the research object. Since innovation performance usually lags, this paper selects 2017 data on the characteristics of corporate executives and 2018 data for innovation performance. Data were selected from the China Stock Market & Accounting Research (CSMAR) database. In order to ensure the feasibility and effectiveness of the study, the initial sample was screened as follows: companies that have suffered losses for more than two consecutive years or are in danger of delisting (i.e., ST and *ST listed companies) and companies with missing data were excluded. Thus, a sample of 634 firms was obtained. SPSS 23.0 was used to analyze the sample data.

### 4.2. Variable Definition and Model Setting

The dependent variable in this paper is enterprise innovation performance. There is no unified measurement index for enterprise innovation performance; thus, this research refers to the approach used by Liu and Zhai (2017) and takes the ratio of enterprises’ research and development investment to total assets as a proxy variable to measure enterprise innovation performance [45]. Independent variables are shallow-factor heterogeneity (i.e., gender heterogeneity, age heterogeneity) and deep-factor heterogeneity (i.e., education background heterogeneity, tenure heterogeneity). The size of the senior management team (i.e., team size) is taken as a control variable for data processing. All variables and specific measurements are shown in Table 1.

The following model is then constructed:(1)$$EIP=\alpha_{0}+\alpha_{1}AH+\alpha_{2}GH+\alpha_{3}EH+\alpha_{4}TH+\alpha_{5}TS+\varepsilon ,$$

Descriptive statistical analysis, correlation analysis, and regression analysis are used to test the hypotheses.

## 5. Results

### 5.1. Descriptive Statistical Analysis

Table 2 shows the descriptive statistical results for each variable. Among the characteristics of the top management team, educational background heterogeneity is the smallest, with a mean value of 0.18 and a standard deviation of 0.04. Gender heterogeneity is the largest, with a maximum value of 1.387 and a minimum value of 0. This indicates that the ratio of males to females in the top management team is unbalanced, with males dominating. The standard deviation of team size is 3.997, with a difference of 33 between the maximum and minimum values, indicating a large gap in team size. This aligns with the large gap in the sizes of GEM-listed enterprises.

### 5.2. Correlation Analysis

Table 3 shows the results of the Pearson and Spearman correlation analyses of all variables in the full sample. It can be seen that enterprise innovation performance is negatively correlated with education background heterogeneity and tenure heterogeneity, but not with other variables. Age heterogeneity is positively correlated with gender heterogeneity and education background heterogeneity but not with other variables. Tenure heterogeneity is negatively correlated with education background heterogeneity and positively correlated with team size, while team size is negatively correlated with education background heterogeneity. Correlation analysis provided preliminary support for Hypothesis 2A and Hypothesis 2B, led to the rejection of Hypothesis 1A and Hypothesis 1B, and showed that there is no influence from team size. In addition, the absolute value of the correlation coefficient of all variables does not exceed 0.4, indicating that there is no serious multicollinearity problem between variables.

### 5.3. Top Management Team Heterogeneity and Enterprise Innovation Performance

In order to better compare and analyze the relative influence of the variables under consideration, this paper tested the relationship between top management team heterogeneity and enterprise innovation performance using six models: (1) Model 1 contains only the control variable, team size, (2) Model 2 contains team size and gender heterogeneity, (3) Model 3 contains team size and age heterogeneity, (4) Model 4 contains team size and education background heterogeneity, (5) Model 5 contains team size and tenure heterogeneity, and (6) Model 6 contains all variables. The regression analysis results of the above six test models are shown in Table 4.

As shown in Table 4, the influence of the team size on enterprise innovation performance is not significant. The main effect regression analysis results of Models 2 to 6 show that age heterogeneity and gender heterogeneity have no significant influence on enterprise innovation performance, while education background heterogeneity and tenure heterogeneity have a significant negative influence thereon. Thus, Hypothesis 1 is rejected, but Hypothesis 2 is supported.

## 6. Discussion and Implications

### 6.1. Discussion

Based on upper echelons theory, this paper examined the influence of background heterogeneity (divided into shallow factors and deep factors) of top management teams on their innovation performance. Based on data of GEM-listed companies from 2016 to 2018, we found that there is no direct correlation between shallow factors and GEM-listed enterprises’ innovation performance, while the heterogeneity of deep factors negatively affects innovation performance. These findings refine research on the influence of heterogeneity of top management team background characteristics on enterprise performance, especially innovation performance, thereby enriching upper echelons theory.

The results regarding shallow factors have nothing to do with enterprise innovation performance, which differs from information decision theory. This may be because the influence of two different directions exists simultaneously, while in the entrepreneurial enterprise environment, the influence of two opposite directions is balanced. Therefore, start-ups in the process of building their top management team do not need to consider the gender ratio and age characteristics of the top management team. Another possibility is that the influence of superficial factors on enterprise innovation performance has an indirect influence on GEM enterprises and must be realized through intermediary mechanisms.

At the same time, Hypothesis 2, regarding the heterogeneity of deep factors of the senior management team, including the heterogeneity of tenure and educational background, was supported. This provides strong evidence to refute the suggestion that the heterogeneity of deep factors positively affects innovation performance. This may be because homogeneous education background and tenure can effectively reduce the communication cost by enhancing the cohesion, identity, and unification of goals, which enables the company’s innovation goals to be accomplished more easily.

### 6.2. Implications

Our findings have several implications. First, attention needs to be paid to differences in the backgrounds of members of the top management team. As the development decisions of a company are closely related to these backgrounds, enterprises should pay attention to them in the process of forming the executive team. Specifically, firms should not only consider whether individuals meet the needs of the company but also take into account the heterogeneity and homogeneity between team members so that they can better work together to drive innovation.

Second, in innovation-oriented enterprises, top management teams should be formed without consideration for gender and age stereotypes or preferences. The heterogeneity of shallow factors is not directly related to the innovation performance of GEM-listed enterprises and, in fact, provides favorable conditions for companies to create a level playing field in the workplace. This suggests that when hiring executives, SMEs should consider the candidate’s ability, character, and suitability for the job, without considering the gender and age composition of the team. This will help to reduce the pressure of assessment on the human resources department and contribute to a better-organized internal promotion mechanism.

Finally, when assembling a top management team, it is important to minimize the heterogeneity of deep factors among team members. This is primarily because, in GEM-listed firms, deep factors harm the innovation performance of the firm. Thus, trying to ensure the consistency of deep elements within the top management team, such as extending the tenure of these team members and hiring managers who have similar educational backgrounds, can be important for team stability. This can improve communication and cooperation between team members, contributing to greater team cohesion, which in turn improves the efficiency and quality of decision-making and thus the long-term development of the business.

## 7. Limitations and Future Research

This paper suffers from several shortcomings, which provide directions for future research. First, this paper studied GEM-listed enterprises as a whole; however, differences across industries may lead to different results. Thus, future research could take into account the characteristics of different industries and carry out a more specific analysis of representative industries. Second, this paper only studied the direct influence of heterogeneity without discussing the mechanism of action of this influence—this should be discussed in the future. Third, the influence of shallow factors on enterprise innovation performance identified in this paper differed from that found in previous studies; future research can examine the reasons for this divergence.

## Figures and Tables

**Table 1 behavsci-12-00164-t001:** Variable definitions and descriptions.

Type	Name	Abbreviation	Definition
Dependent Variable	Enterprise innovation performance	*EIP*	Research and development investment/total assets
Independent Variable	Gender heterogeneity	*GH*	The standard deviation of gender/mean of gender; mean of gender = total gender/total number of top management team members
	Age heterogeneity	*AH*	The standard deviation of age/mean of age; mean of age = total age/total number of top management team members
	Education background heterogeneity	*EH*	The standard deviation of educational background/mean of educational background; mean of educational background = sum of class assignment/total number of the top management team in one enterprise
			The value of high school, technical secondary school or below is 0; junior college is 1, undergraduate degree is 2; master’s degree is 3; doctor ate is 4
	Tenure heterogeneity	*TH*	The standard deviation of tenure/mean of tenure; tenure = sum of executive’s current duties/total number of top management team members
Control Variable	Team size	*TS*	Number of top management team members

**Table 2 behavsci-12-00164-t002:** Descriptive statistical analysis.

Variable	Mean	Standard Deviation	Minimum	Maximum
EIP	0.034	0.024	0.001	0.171
GH	0.522	0.212	0.000	1.387
AH	0.179	0.043	0.000	0.327
EH	0.309	0.095	0.000	0.717
TH	0.529	0.269	0.000	1.210
TS	15.913	3.997	1.000	34.000

**Table 3 behavsci-12-00164-t003:** Pearson and Spearman correlation coefficients.

	EIP	GH	AH	EH	TH	TS
EIP	1.000	0.032	−0.038	−0.174 **	−0.145 **	−0.003
GH	0.029	1.000	0.210 **	−0.013	0.010	−0.054
AH	−0.060	0.187 **	1.000	0.085 *	0.051	0.011
EH	−0.149 **	0.003	0.078 *	1.000	−0.105 **	−0.083 *
TH	−0.174 **	0.009	0.074	−0.085 *	1.000	0.354 **
TS	−0.021	−0.099 *	−0.027	−0.072	0.340 **	1.000

**Note:** The table is divided into two parts, with 1.000 as the diagonal. Spearman’s correlation coefficient is at the bottom left, and Pearson’s correlation coefficient is at the top right. ** and * are significant at the level of 5% and 10%, respectively.

**Table 4 behavsci-12-00164-t004:** Regression analysis results.

Variable	Model 1	Model 2	Model 3	Model 4	Model 5	Model 6
Constant	0.034 ***	0.032 ***	0.038 ***	0.050 ***	0.037 ***	0.053 ***
TS	0.000	0.000	0.000	0.000	0.000	0.000
GH		0.004				0.004
AH			−0.021			−0.012
EH				−0.045 ***		−0.047 ***
TH					−0.015 ***	−0.016 ***
R^2^	0.000	0.001	0.001	0.030	0.021	0.060
Adjusted R^2^	−0.002	−0.002	−0.002	0.027	0.024	0.053
F-statistic	0.005	0.318	0.447	9.902 ***	7.679 ***	8.081 ***

**Note:** *** is significant at the level of 1%.

## Data Availability

The data analyzed in this paper are proprietary and therefore cannot be posted online.

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
