# Peer review of "Is Heterogeneity Better? The Impact of Top Management Team Characteristics on Enterprise Innovation Performance"

_behavsci, 2022, doi:10.3390/bs12060164_

Round 1
Reviewer 1 Report
Concise, explicit, and well written paper. In formal and/or technical terms, the paper is correct. It shows a potential use for social engineering and the instrumentalization of the human being, which should be questioned.
General considerations
Lines 104-107 and 276: I cannot agree that gender, race, age, nationality, career experience, educational background, industry status or tenure are considered personality variables/traits. They are indeed demographic variables. There are in literature several personality variables (temperament, convictions, attitudes, dispositions etc.) but different from those addressed by authors. The same question arises in line 287, since the authors do not address the human character as an element/variable of their investigation before.
Lines 109-111: It seems that it would be helpful for the reader if authors explained a little more the criteria that allow distinguishing between shallow and deep factors.
Specific considerations
Line 66: Unfinished sentence: “…in strategic decisions and [16].”
Line 135: Correct the sentence: “…age heterogeneity means equates to…”.
Lines 182-183: Unexplained abbreviation “ST and *ST”.
Lines 189-191 and 400-401: It seems to be an inconsistency here because the authors refer to “Liu and Qu (2017)”, while in the References they mention “38. Liu, Y; Zhai,H.”.
Line 367: It should be reviewed “2015, 2015, 16695-16695”.
Line 381: Add “l” in the word “Journa”.
It seems that authors should add doi in the References.
Author Response
Thank you very much for taking the time to review this manuscript again. We appreciate all your comments and suggestions! Based on the instructions provided in your letter, we uploaded the file of the revised manuscript (the revised part using the“Track Changes” function in the manuscript). Please find the itemized responses below and the revisions/corrections in the re-submitted files.
Reviewers’ comments and the explanations of improvement:
|
Reviewer: |
Responses |
|
Lines 104-107 and 276: I cannot agree that gender, race, age, nationality, career experience, educational background, industry status or tenure are considered personality variables/traits. They are indeed demographic variables. There are in literature several personality variables (temperament, convictions, attitudes, dispositions etc.) but different from those addressed by authors. The same question arises in line 287, since the authors do not address the human character as an element/variable of their investigation before. |
Thank you so much for your approval and suggestions.
We apologize for the misuse of the term “personality variables/traits” and revise them to “background” in this turn, based on previous research e.g., Zimmerman (2010).
Zimmerman, M.A. The influence of top management team heterogeneity on the capital raised through an initial public offering. Entrepreneurship Theory & Practice 2010, 32, 391-414. DOI: https://doi.org/10.1111/j.1540-6520.2008.00233.x.
|
|
Lines 109-111: It seems that it would be helpful for the reader if authors explained a little more the criteria that allow distinguishing between shallow and deep factors. |
Thanks for your advice. We add more criteria as follows: To better explain the divergent findings, top management team backgrounds have been divided into shallow factors, which include some factors that are predetermined by birth and relatively difficult to change, such as gender, race, age, nationality, etc.; and deep factors, including career experience, education background, industry status, tenure, and other variables that are more dependent on acquired experience [33]. Specifically, shallow factors in team members refer to differences based on some demographic variables, while deep factors are formed by the past learning and work experiences of team members [34]. |
|
Line 66: Unfinished sentence: “…in strategic decisions and [16].” |
Thank you for pointing this out.
We complete our expression as follows: “Furthermore, researchers added to and revised the upper echelons theory, arguing that the top management team comprises the individuals that are most deeply involved in decision-making and -execution, and beginning to pay more attention to the important role of the top management team in strategic decisions and performance improvements[18-20].” |
|
Line 135: Correct the sentence: “…age heterogeneity means equates to…”. |
Thank you for pointing this out.
We revise our expression as follows: “Under such conditions, age heterogeneity means a wider range of intellectual assets, richer information sources, and a stronger ability to solve new problems compared to the case in more homogeneous teams [42].” |
|
Lines 182-183: Unexplained abbreviation “ST and *ST”. |
Thanks for your advice.
We are sorry for the confusion by our poor expression and revise it as follows: In order to ensure the feasibility and effectiveness of the study, the initial sample was screened as follows: companies that have suffered losses for more than two consecutive years or are in danger of delisting, i.e., ST and *ST listed companies, of which those with missing data were excluded. |
|
Lines 189-191 and 400-401: It seems to be an inconsistency here because the authors refer to “Liu and Qu (2017)”, while in the References they mention “38. Liu, Y; Zhai,H.”. |
Thank you for pointing this out.
Qu is a polyphonic character and should be pronounced "Zhai" when used as a surname. We apologize for our carelessness and change “Liu and Qu (2017)” to “Liu and Zhai (2017)”. |
|
Line 367: It should be reviewed “2015, 2015, 16695-16695”. |
Thank you for pointing this out.
We are sorry for our wrong reference format and revise it as follows: Gschmack, S.; Reimer, M.; Schäffer, U. TMT heterogeneity and strategic decision quality. In Academy of Management Proceedings, Publisher: Published Online 2017, Volume 2015, pp. 16695-16695. |
|
Line 381: Add “l” in the word “Journa”. |
Thank you for pointing this out and we complete our spelling. |
|
It seems that authors should add doi in the References. |
Thank you for pointing this out.
We add DOI in the References follow your advice. |

Reviewer 2 Report
Dear Authors,
This paper is very well written. I like the paper and thank you for the opportunity to read yours work. This article is of extremely high scholarly standards, it is original in numerous respects and certainly most timely in terms of Behavioral Sciences relevance.
Best regards!
Author Response
Thank you for your approval of our work.
Best regards!
Reviewer 3 Report
This paper considers the influence of top management team characteristics’ heterogeneity on GEM-listed enterprises’ innovation performance. From the perspective of optimizing top management team human capital and improving enterprise innovation performance. The results show, and the authors conclude, that the effect of shallow-factor heterogeneity of the top management team on enterprise innovation performance is not significant, but deep-factor heterogeneity has a negative effect. According to the authors, these findings indicate that deep-factor heterogeneity is more important than shallow-factor heterogeneity and be subject to closer attention. According to the authors, if a GEM-listed enterprise wants to set up a strong top management team that contributes to the improvement of enterprise innovation performance, it must examine the consistency of each member’s educational background and extend their tenure as long as possible without considering the gender ratio or age structure of the team.
The study presents a good sample for the empirical part. The study's conclusions are objective.
The work is well structured but presents outdated bibliography. More than 80% of the authors cited are publications over 12 years old. Authors should increase citation of more recent authors. Typically, 50% of the authors cited are from the last 5 years. Specifically, of the 38 authors cited, only 11 are published after 2015!
Given that the bibliography used must be more recent, the text must be improved. It is incorrect for a paper from the year 2022 to present 23.68% of the bibliography cited from the last century. This must be improved and changed.
Author Response
Thank you very much for taking the time to review this manuscript again. We appreciate all your comments and suggestions! Based on the instructions provided in your letter, we uploaded the file of the revised manuscript (the “track changes” function was used to make revisions in the manuscript). Please find the itemized responses below and the revisions/corrections in the re-submitted files.
Reviewers’ comments and the explanations of improvement:
|
Reviewer: |
Responses |
|
The work is well structured but presents outdated bibliography. More than 80% of the authors cited are publications over 12 years old. Authors should increase citation of more recent authors. Typically, 50% of the authors cited are from the last 5 years. Specifically, of the 38 authors cited, only 11 are published after 2015! Given that the bibliography used must be more recent, the text must be improved. It is incorrect for a paper from the year 2022 to present 23.68% of the bibliography cited from the last century. This must be improved and changed. |
Thank you so much for your approval and suggestions.
In response to your valuable comments, we replace 5 outdated papers with literature published after 2015, in addition to adding 8 more papers published after 2015. Further details can be found in the references section of the manuscript (or you can see these part as follows).
|
Or you can choose to view the modified version as follows:
References
- Su, Z., et al. Top management team’s participative decision-making, heterogeneity, and management innovation: an information processing perspective. Asia Pacific Journal of Management 2022. 39, 149-171. DOI: 10.1007/s10490-021-09752-2.
- Yu, C., et al., Do top management teams’ expectations and support drive management innovation in small and medium-sized enterprises? Journal of Business Research 2022. 142, 88-99. DOI: https://doi.org/10.1016/j.jbusres.2021.12.035.
- Wang, H.; Zhou, Z.; Pan, H. The Influence of the Reputation of Finance Intermediaries on IPO Underpricing in China Growth Enterprize Markets (GEM). Management Science & Engineering 2012, 6, 549-58. DOI: 3968/j.mse.1913035X20120601.2660
- Chu, C.-C., et al. Uncertainty, venture capital and entrepreneurial enterprise innovation—Evidence from companies listed on China's GEM. Pacific-Basin Finance Journal 2021, 68, 101576. DOI: https://doi.org/10.1016/j.pacfin.2021.101576.
- Hambrick, D.C.; Mason, P.A. Upper Echelons: The Organization as a Reflection of Its Top Managers. Academy of Management Review 1984, 9, 193-206. DOI: https://doi.org/10.5465/amr.1984.4277628.
- Gil, D.; Hartmann, F.; Maas, V. Top Management Team Heterogeneity, Strategic Change and Operational Performance*. British Journal of Management 2008, 19, 222-234. DOI: 10.1111/j.1467-8551.2007.00545.x.
- Janani, S., et al. Marketing experience of CEOs and corporate social performance. Journal of the Academy of Marketing Science, 2022 50, 460-481. DOI: 10.1007/s11747-021-00824-9.
- Xia, T.; Liu, X. The innovation paradox of TMT political capital in transition economy firms. Journal of Business Research 2022 142, 775-790. DOI: https://doi.org/10.1016/j.jbusres.2022.01.011.
- Pitcher, P.; Smith, A. Top Management Team Heterogeneity: Personality, Power, and Proxies. Organization Science 2001, 12, 1-18. DOI: 10.1287/orsc.12.1.1.10120.
- Nasta, L.; Pirolo, L.; Wikström, P. Diversity in creative teams: a theoretical framework and a research methodology for the analysis of the music industry. Creative Industries Journal 2016, 9, 97-106. DOI: 10.1080/17510694.2016.1154653.
- Harvey, S. A different perspective: The multiple effects of deep level diversity on group creativity. Journal of Experimental Social Psychology 2013, 49, 822-832. DOI: https://doi.org/10.1016/j.jesp.2013.04.004.
- Shin, S.J., et al. Cognitive Team Diversity and Individual Team Member Creativity: A Cross-Level Interaction. Academy of Management Journal 2012, 55, 197-212. DOI: 10.5465/amj.2010.0270.
- White, J.V.; Borgholthaus, C.J. Who’s in charge here? A bibliometric analysis of upper echelons research. Journal of Business Research 2022, 139, 1012-1025. DOI: https://doi.org/10.1016/j.jbusres.2021.10.028.
- Kumar, A. Leadership and decision-making: Top management team age demographic and environmental strategy. Journal of Management & Organization 2020, 1-17. DOI: 10.1017/jmo.2019.91.
- Narayan, S., Sidhu, J.S. and Volberda, H.W. From attention to action: The influence of cognitive and ideological diversity in top management teams on business model innov Journal of Management Studies, 2021, 58, 2082-2110.DOI: https://doi.org/10.1111/joms.12668.
- Parola, H.R.; Ellis, K.M.; Golden, P. Performance effects of top management team gender diversity during the merger and acquisition process. Management Decision 2015, 53, 57-74. DOI: 10.1108/MD-03-2014-0141.
- Wu, J., et al. The performance impact of gender diversity in the top management team and board of directors: A multiteam systems approach. Human Resource Management 2022, 61, 157-180. DOI: https://doi.org/10.1002/hrm.22086.
- Carpenter, M.A.; Geletkanycz, M.A.; Sanders, W.G. Upper Echelons Research Revisited: Antecedents, Elements, and Consequences of Top Management Team Composition. Journal of Management 2004, 30, 749-778. DOI: https://doi.org/10.1016/j.jm.2004.06.001.
- Azam, A., et al. Top management team international experience, international information acquisition and international strategic decision rationality. Review of International Business and Strategy 2020, 30, 441-456. DOI: 10.1108/RIBS-01-2020-0010.
- Ugalde Vásquez, A.F. and Naranjo-Gil, Management Accounting Systems, Top Management Teams, and Sustainable Knowledge Acquisition: Effects on Performance. Sustainability 2020, 12, 1-14. DOI: https://doi.org/10.3390/su12052132.
- Naranjo-Gil, D. How management teams use information and control systems to manage hospitals. Gaceta sanitaria 2016, 30, 287-292. DOI: 10.1016/j.gaceta.2015.12.009.
- Wang, X.; Deng, S.; Alon, I. Women executives and financing pecking order of GEM-listed companies: Moderating roles of social capital and regional institutional environment. Journal of Business Research 2021, 136, 466-478. DOI: https://doi.org/10.1016/j.jbusres.2021.07.055.
- Sahaym, A., et al. Mixed blessings: How top management team heterogeneity and governance structure influence the use of corporate venture capital by post-IPO firms. Journal of Business Research 2016, 69, 1208-1218. DOI: https://doi.org/10.1016/j.jbusres.2015.09.012.
- Zhou, Y., et al. Effects of Top Management Team Characteristics on Patent Strategic Change and Firm Performance. Frontiers in Psychology, 2022, 12. DOI: 10.3389/fpsyg.2021.762499.
- Gschmack, S.; Reimer, M.; Schäffer, U. TMT heterogeneity and strategic decision quality. In Academy of Management Proceedings, Publisher: Published Online 2015, Volume 2015, pp. 16695-16695. DOI: 10.5465/AMBPP.2015.16695abstract.
- Donaldson, L. Strategic Leadership: Top Executives and Their Effects on Organizations. Australian Journal of Management 1997, 22, 221-224. DOI: 10.1177/031289629702200205.
- Wu, W.Y.; Chiang, C.Y.; Jiang, J.S. Interrelationships between TMT management styles and organizational innovation. Industrial Management & Data Systems 2002, 102, 171-183. DOI: 10.1108/02635570210421363.
- Zenger, T. Organizational Demography: The Differential Effects of Age and Tenure Distributions on Technical Communication. The Academy of Management Journal 1989, 32, 353-376. DOI: 10.5465/256366.
- Alexiev, A.S., et al. Top Management Team Advice Seeking and Exploratory Innovation: The Moderating Role of TMT Heterogeneity. Journal of Management Studies 2010, 47, 1343-1364. DOI: https://doi.org/10.1111/j.1467-6486.2010.00919.x.
- Knight, D., et al. Top management team diversity, group process, and strategic consensus. Strategic Management Journal 1999, 20, 445-465. DOI: https://doi.org/10.1002/(SICI)1097-0266(199905)20:5<445::AID-SMJ27>3.0.CO;2-V.
- Ndofor, H.A.; Sirmon, D.G.; He, X. Utilizing the firm's resources: How TMT heterogeneity and resulting faultlines affect TMT tasks. Strategic Management Journal 2015, 36, 1656-1674. DOI: https://doi.org/10.1002/smj.2304.
- Mehrabi, H., Coviello, N. and Ranaweera, C. When is top management team heterogeneity beneficial for product exploration? Understanding the role of institutional pressures. Journal of Business Research 2021, 132, 775-786. DOI: https://doi.org/10.1016/j.jbusres.2020.10.057.
- Simons, T.; Pelled, L.H.; Smith, K.A. Making Use of Difference: Diversity, Debate, and Decision Comprehensiveness in Top Management Teams. Academy of Management Journal 1999, 42, 662-673. DOI: 10.5465/256987.
- Su, K.; Kang, S. and Tao, X. Executive characteristics, absorptive capacity constructs and innovation performance - an analysis based on the degree of variation in executive characteristics.Enterprise Economy 2020, 39, 60-67. (in Chinese) DOI: 10.13529/j.cnki.enterprise.economy.2020.02.008
- Richard, O.C.; Shelor, R.M. Linking top management team age heterogeneity to firm performance: juxtaposing two mid-range theories. The International Journal of Human Resource Management 2002, 13, 958-974. DOI: 10.1080/09585190210134309.
- Li, P.-Y. The impact of the top management teams’ knowledge and experience on strategic decisions and performance. Journal of Management & Organization 2017, 23, 504-523. DOI: 10.1017/jmo.2016.24.
- Carpenter, M.A. The implications of strategy and social context for the relationship between top management team heterogeneity and firm performance. Strategic Management Journal 2002, 23, 275-284. DOI: https://doi.org/10.1002/smj.226.
- Post, C., Lokshin, B. and Boone, What Changes after Women Enter Top Management Teams? A Gender-Based Model of Strategic Renewal. Academy of Management Journal 2022, 65, 273-303. DOI: 10.5465/amj.2018.1039.
- Yasser, Q.R., et al. Corporate social responsibility and age of productivity: A study on emerging economies. Thunderbird International Business Review 2020, 62, 661-674.DOI: https://doi.org/10.1002/tie.22172.
- Adams, R.B.; Ferreira, D. Women in the boardroom and their impact on governance and performance. Journal of Financial Economics 2009, 94, 291-309. DOI: https://doi.org/10.1016/j.jfineco.2008.10.007.
- Peng, W.; Duan, X. Analysis on Growth Vulnerability of GEM Listed Enterprises. Science and Technology Management Research, 2013, 33, 206-209. (in Chinese). DOI: 10.3969/j.issn.1000-7695.2013.16.044.
- Amason, A.C. Distinguishing the Effects of Functional and Dysfunctional Conflict on Strategic Decision Making: Resolving a Paradox for Top Management Teams. Academy of Management Journal 1996, 39, 123-148. DOI:10.2307/256633.
- Smith, K.G., et al. Top Management Team Demography and Process: The Role of Social Integration and Communication. Administrative Science Quarterly 1994, 39, 412-438. DOI: 10.2307/2393297.
- McDonald, M.L.; Westphal, J.D. Getting by with the advice of their friends: CEOs' advice networks and firms' strategic responses to poor performance. Administrative science quarterly 2003, 48, 1-32. DOI: https://ssrn.com/abstract=936733.
- Liu, Y.; Zhai, H. Research on Vertical Dyad Linkage in Top Management Team and Corporate Innovation. Science & Technology Progress and Policy 2017, 34, 104-111. (in Chinese). DOI: 10.6049/kjjbydc.2016090274.
